# Wait, Where’s the Flynn Effect on the WAIS-5?

**DOI:** 10.3390/jintelligence12110118

**Published:** 2024-11-15

**Authors:** Emily L. Winter, Sierra M. Trudel, Alan S. Kaufman

**Affiliations:** 1School of Health Sciences, Touro University, 3 Times Square, New York, NY 10036, USA; 2Department of Psychology, Marist College, 3399 North Road, Poughkeepsie, NY 12601, USA; sierra.trudel@marist.edu; 3Neag School of Education, University of Connecticut, Charles B. Gentry Building, 249 Glenbrook Road U-3064, Storrs, CT 06269, USA; alan.kaufman@uconn.edu

**Keywords:** Flynn effect, IQ tests, cognitive assessment, Wechsler scales, WAIS-5, WAIS-IV, intelligence testing, capital punishment cases, intellectual disabilities

## Abstract

The recent release of the WAIS-5, a decade and a half after its predecessor, the WAIS-IV, raises immediate questions about the Flynn effect (FE). Does the traditional FE of points per decade in the U.S. for children and adults, identified for the Full Scale IQs of all Wechsler scales and for other global IQ scores as well, persist into the 2020s? The WAIS-5 Technical and Interpretive Manual provides two counterbalanced validity studies that address the Flynn effect directly—*N* = 186 adolescents and adults (16–90 years, mean age = 47.8) tested on the WAIS-IV and WAIS-5; and *N* = 98 16-year-olds tested on the WISC-V and WAIS-5. The FE is incorporated into the diagnostic criteria for intellectual disabilities by the American Association on Intellectual and Developmental Disabilities (AAIDD), by DSM-5-TR, and in capital punishment cases. The unexpected result of the two counterbalanced studies was a reduction in the Flynn effect from the expected value of 3 IQ points to 1.2 points. These findings raise interesting questions regarding whether the three point adjustment to FSIQs should be continued for intellectual disability diagnosis and whether the federal courts should rethink its guidelines for capital punishment cases and other instances of high stakes decision-making. Limitations include a lack of generalization to children, the impact of the practice effects, and a small sample size.

## 1. Introduction

### 1.1. Intelligence

Broadly defined, intelligence is a person’s ability to learn from their experience, adapt, influence, and choose environments ([21]; [44]). Several theories seek to understand the underlying makeup and components of intelligence ([44]). Intelligence theories consider the biological basis for intelligence ([8]; [23]), understanding the systems of the mind ([7]; [43]), and psychometric evidence ([4]). At the heart of intelligence is an individual’s problem-solving skills and capacity for knowledge acquisition ([19]; [51]). Domains within intelligence include areas such as fluid reasoning, visual-spatial processing, working memory, processing speed ([41]), attention, simultaneous and successive processing, and planning ([6]), among others. Together, these domains can influence an individual’s ability to think, act, and learn within their environment ([44]).

Despite the importance of the measurement of separate cognitive abilities and processes, and an overwhelming research base that supports the validity and utility of these theories ([13]), the global score (*g*) often assumes prominence for decision-making. Wechsler’s scales have been the most popular IQ tests worldwide for more than a half century (e.g., [26]; [33]; [63]), and Wechsler’s prominence shows no sign of subsiding with the publication of the state-of-the-art Wechsler Adult Intelligence Scale—Fifth Edition (WAIS-5; [58], [59]). The WAIS-5 yields scales that are easily interpretable from the vantage point of Cattell–Horn–Carroll (CHC) theory, and it includes new subtests that have emerged from theories of working memory and processing speed; but, ultimately, it is the *g* score—Full Scale IQ (FSIQ)—that is of immediate extraordinary interest to clinicians and researchers alike because of its central role in the diagnosis of intellectual disabilities (ID). After a wait of 16 years, longer than expected for a Wechsler scale due to the COVID-19 pandemic, one question that needs to be answered concerns the Flynn effect.

### 1.2. Flynn Effect

Cross-culturally, shifts in cognitive intelligence (as measured by IQ) have been observed since 1932 ([37]), through a phenomenon that was later dubbed the Flynn effect in tribute to the researcher who introduced the concept more than a generation ago ([15], [16], [17], [14]). These changes in population-wide intelligence range in their magnitude depending on the country (gains in some nations average 7 or 8 points per decade), with the United States reporting a growth of 3 IQ points every ten years, a pattern that has been constant for about a century ([15], [16], [17]; [30]; [48]), with the effect potentially observed in animal populations as well ([62]). Simply put, the Flynn effect suggests that younger generations score higher on IQ tests than those prior ([11]). This phenomenon has extended to other clinical areas, including in academic skills, such as vocabulary and math scores ([32]) or cognitive subskills, such as increases in abstract reasoning, verbal skills, and vocabulary, while decreasing rote calculation, visual spatial abilities, and verbal meanings ([5]). Dworak and colleagues (2023) remind us that the Flynn effect does not mean that younger generations are practically “smarter”, but rather they are outperforming their older colleagues on similar standardized measures, thus with measurements favoring those who are younger.

The Flynn effect is a population-based observation; however, this phenomenon impacts groups of people and individuals in “high stakes” decision-making, namely in the diagnosis of intellectual disability (ID), in guiding educational placement and governmental services, and in capital punishment cases ([27]; [39]). Despite their ubiquitous use in such decision-making, [17] ([17]) warned that these effects would not be forever reaching, but rather, for younger groups, the effects he hypothesized would diminish as time progressed. Recent research, along with findings from this paper, highlight that Flynn’s original hypothesis may indeed be correct that the effects may not last forever.

For the first time in history, recent research has suggested a plateau of this effect, and in some cases a reverse Flynn effect cross-culturally, which has already been observed in several European countries ([3]; [9]; [10]; [45]; [46], [47]). Yet, it is important to note, despite the novelty of these findings, many studies espouse considerable methodological limitations, such as through the use of unconventional “intelligence” measurements (or using other types of skills as a “proxy” for IQ; [42]), relying on group-administered assessments instead of individual assessments, small sample sizes, or using abbreviated (instead of Full Scale) measures ([31]). Even within counterbalanced research (such as within the present paper), concerns with sample size and practice effects, as well as generalizability (such as with differing age groups), may come into effect.

Other scholars argue that although the Flynn effect may be alive and well, the effect’s presence may be inconsistent, varying depending on age or range of intelligence. For instance, on a Wechsler measure, the Flynn effect was observed only for children in the low IQ ranges ([25]), as children (when using the new norms of the test) were more likely to receive an educational classification of intellectual disability as compared to the prior version (i.e., suggesting that with newer, more stringent, norms, students were “easier” to classify). These studies have also faced design flaws with smaller sample sizes ([64]) and extrapolating findings from different educational classifications, such as learning disabilities ([38]). A more recent study on the Flynn effect, using an abbreviated measure of *g* (matrices) with more than 10,000 adolescents aged 13–18, found the Flynn effect within particular segments of the populations; the Flynn effect was a significant function of both age and ability level. Specifically, Platt and colleagues (2019) observed that IQ rose about 3.5 points per decade for individuals with an IQ greater than 130 (consistent with Flynn effect predictions) but decreased 4.9 points for people with IQs under 70 (i.e., a reverse Flynn effect). Similarly, chronological age interacted significantly with the direction and magnitude of the Flynn effect. At age 13, IQ rose by 2.3 points, consistent with a Flynn effect, but IQ decreased by 1.6 points at age 18, a reverse Flynn effect.

These findings, across multiple studies, suggest that generalizing the Flynn effect to all people within the population may not be an accurate reflection of the phenomenon; rather, the effects are more nuanced based on ability and age, amongst other factors. It has been widely known from Flynn’s early publications that nations vary widely in the magnitude of generational change in IQ, and this finding has been confirmed by meta-analysis ([30]). In the United States, “Flynn’s early publications indicated that children and adults score higher on IQ tests than previous generations at the rate of approximately 3 IQ points per decade”; ([31]) this finding also has also been verified by meta-analysis ([30]), although the value is sometimes closer to 2 points ([48]; [60]). [48] ([48]) noted a rise in IQ with an average of 2.31 points per ten years (*SD* = 0.15) in the United States, observing consistency across various factors, including age and ability level; [60] ([60]) studied the Flynn effect cross-culturally, including within the United States, and noted in their meta-analytic review that the effect was most present in middle-income countries and younger generations (an average gain of +0.22 points per year). Cross-cultural research also suggests that positive Flynn effects may be occurring in less economically developed nations, whereas negative Flynn effects are observed in economically advantaged countries, revealing an economic and cognitive convergence ([28]).

### 1.3. Why Does the Flynn Effect Matter?

For adult populations, the Flynn effect matters substantially, especially for those with ID in high stakes decision-making contexts. In the Diagnostic and Statistical Manual of Mental Disorders, Fifth Edition, Text Revision (DSM-5-TR; [1]), an intellectual disability is marked by deficits in adaptive functioning and “deficits in intellectual functioning, such as reasoning, problem-solving, planning, abstract thinking, judgment, academic learning, and learning from experience, confirmed by both clinical assessment and individualized, standardized intelligence testing” (p. 37). Individuals with intellectual disabilities typically perform at least two standard deviations below the mean (e.g., 65–75 which includes ±5 points for measurement error). Furthermore, the DSM-5-TR recommends taking into consideration the Flynn effect as one of the key factors that impact test scores ([1]).

Additionally, the American Association on Intellectual and Developmental Disabilities (AAIDD) stipulates that IQs must be adjusted for the Flynn effect when diagnosing intellectual disabilities: “Current best practice guidelines recommend that in cases in which an IQ test with aged norms is used as part of a diagnosis of ID, a correction of the full-scale IQ score of 0.3 points per year since the test norms were collected is warranted” ([40]).

IQ adjustments for ID play a critical role in the criminal justice system and, more specifically, regarding the death penalty ([27]). The Flynn effect has had an impact on capital punishment cases for individuals having a suspected intellectual disability. The outcome from the 2002 U.S. Supreme Court case [2] ([2]) invoked the Eighth Amendment to prohibit the execution of individuals with an intellectual disability—such an execution would amount to “cruel and unusual punishment”. Since this ruling, federal and state courts have typically applied the Flynn effect in cases where the defendant is suspected of having an intellectual disability to mitigate the irrevocable punishment of the death penalty ([12]). Additionally, Reynolds and colleagues (2010) note that the inclusion of the Flynn effect on capital cases involving the death penalty is a “true matter of life and death” (p. 477). The authors implore, “To do less is to do wrong—what possible justification could there be for issuing estimates of general intelligence in a death penalty case that are less than the most accurate estimates obtainable?” ([34]).

In the school and clinical settings, the Flynn effect adjustment is less relevant. [27] ([27]) states, “The use of the Flynn effect correction in clinical settings is less of an issue given that psychologists in such settings typically have more leeway to interpret scores as ranges, invoke clinical judgment, and incorporate information regarding measurement error in interpretation of scores when making a diagnosis” (p. 160). School psychologists follow similar guidelines within the school setting. Additionally, in schools, the Flynn effect is less relevant due to the evaluation mandates put forth by the Individuals with Disabilities Education Act ([22]). Students who qualify for special education are required to be re-evaluated at least once every three years ([22]). However, for students with an intellectual disability who are transitioning out of high school (i.e., over the age of 16), obsolete tests and outdated norms for adult cognitive assessment tools can lead to inflated IQ scores, which may change eligibility classification from ID to learning disability ([48]). For students who are leaving high school and entering the independence required of adulthood, this change in classification may impact government supported social security and disability supports that require evidence of impaired intellectual functioning ([48]).

### 1.4. Statement of the Problem

Intelligence tests for children and adults are traditionally re-normed and released with contemporary normative data every 10–15 years to update content and make other modifications based on societal changes, culture shifts, advances in theory, and innovations in technology. But the urgency to revise and re-standardize became paramount when the Flynn effect became an axiomatic fact of life, sometime in the 1990s. The 1949 WISC was not revised for a quarter century ([52], [53]) and the same was true for the WAIS and WAIS-R ([55], [56]). Test publishers are no longer able to get away with that slow pace. Clinicians were able to quote chapter and verse of the Flynn effect and demand accountability from test publishers. They became well aware that IQ test norms become out of date at the rate of three IQ points per decade. Older norms tend to produce spuriously high IQs because they become increasingly “soft” as time goes by (i.e., children and adults answer more questions correctly, and solve more problems accurately, than their parents did). The artificially inflated IQs require adjustment when diagnosing ID, namely 0.3 points must be subtracted from a person’s Wechsler FSIQ for each year that its norms are out of date ([40]). Test publishers listened, with a chief impetus coming from the proliferation of “Atkins cases” following the 2002 court ruling.

The WAIS-IV, normed in 2007–2008, yields IQs that are spuriously high by about 5 points. Although ID diagnoses are dependent on adaptive behavior as well as IQ, it is quite clear that subtracting 5 points from an incarcerated person on death row’s FSIQ can truly be a matter of life and death. The publication of the WAIS-5 in August 2024 ([58], [59]) immediately raised the question of whether the traditional 3-points-per-decade Flynn effect has continued into the mid-2020s, or whether it has reduced in magnitude, as it has in some other nations. Validity data presented in the WAIS-5 Technical and Interpretive Manual ([59]) based on 186 adolescents and adults tested on the WAIS-IV and WAIS-5 in counterbalanced order—as well as a second counterbalanced study of 98 16-year-olds tested on the WISC-V and WAIS-5—directly address the magnitude of the Flynn effect for contemporary American society.

### 1.5. Research Questions

What is the magnitude of the Flynn effect for adolescents and adults in post-pandemic American society based on the results of two rigorously conducted validity studies reported in the WAIS-5 Technical and Interpretive Manual ([59])?

What are the implications of the results of this study for ID diagnosis, especially on its potential impact for capital punishment court cases and, in general, on high-stakes decision-making in cognitive assessment?

## 2. Materials and Methods

### 2.1. Participants

As reported by [59] ([59], Table 5.3), data for two samples were analyzed in this investigation and both tested post-pandemic in 2023 during the standardization and validation of the WAIS-5.

The first sample participated in the WAIS-IV—WAIS-5 study: 186 adolescents and adults ages 16–90 (mean age = 47.8); 65.1% female. Ethnicity = 15.6% African American, 10.2% Asian, 22.0% Hispanic, 43.0% White, 9.1% Other. Education = 16.6% less than 12 years schooling, 23.7% high school graduates, 23.1% some college, 37.6% college graduates.

The second sample participated in the WISC-V—WAIS-5 study: 98 16-year-olds (mean age = 16.4); 60.2% female. Ethnicity = 11.2% African American, 3.1% Asian, 23.7% Hispanic, 50.0% White, 2.0% Other. Education = 21.4% less than 12 years schooling, 18.4% high school graduates, 26.5% some college, 33.7% college graduates.

### 2.2. Procedure

As reported by [59] ([59]), 186 adolescents and adults, ages 16–90, were tested in counterbalanced order on the 2008 WAIS-IV and the 2024 WAIS-5. The intervals between tests ranged from 7 to 134 days with a mean of 28.2 days. In a second study, 98 16-year-olds were tested in counterbalanced order on the 2014 WISC-V and the 2024 WAIS-5. The intervals between tests ranged from 1 to 112 days with a mean of 26.9 days. Pearson product moment correlation analyses were conducted for each study to assess the construct validity of the new WAIS-5. Full Scale IQs correlated 0.92 between the two versions of WAIS, and 0.87 between the fifth editions of WISC and WAIS ([58], [59]). Of special interest for our study are the mean FSIQs for the two Wechsler scales in each study. This particular paper reports additional Mean IQ Difference, not reported in the manual, which has implications for understanding the Flynn effect between the WAIS-IV, WISC-V, and WAIS-5.

### 2.3. Instrument

The WAIS-5 ([58], [59]) was standardized between February 2023 and January 2024 on 2020 adolescents and adults aged 16 through 90 years. The sample comprised 180 per age band for ages 16–69 and 100 per age band for ages 70–90. It was stratified by age, sex, ethnicity, education level, and geographic region; proportions of the normative sample matched closely with the proportions reflected in the 2022 U.S. Bureau of the Census data ([36]). Within sex, 16 examinees reported gender as different from their sex (*n* = 6 female for sex and indicated man for gender; *n* = 5 male for sex and indicated woman for gender; *n* = 4 gender as nonbinary; *n* = 1 genderqueer).

The WAIS-5 includes 20 subtests organized into five primary indexes, and a diverse array of clinically meaningful ancillary indexes; it yields four global scores—FSIQ, Nonverbal Index, Nonmotor Index, and General Ability Index. For our study, we focused only on the FSIQ, which is a composite of seven subtests—Vocabulary, Similarities, Block Design, Matrix Reasoning, Figure Weights, Digit Sequencing, and Coding. The mean reliability of the FSIQ averaged 0.97 across the age range; the stability coefficient was 0.93 for 201 individuals aged 16–90 that were tested twice (average interval = 29 days).

## 3. Results

Table 1 presents the results of the present investigation. Analyses from both counterbalanced studies yielded the identical unexpected results—a dramatic, unexpected Flynn effect of 1.2 IQ points per decade, well below the values of 2.6–2.8 points identified more than 15 years ago based on comparable WAIS-III vs. WAIS-IV analyses ([57]; data also presented in Table 1), strikingly lower than the 3 points per decade identified from meta-analyses ([30]; [48]) and sanctioned as an adjustment to be made for Wechsler FSIQs in ID diagnosis by AAIDD ([40]), especially in capital punishment court cases.

The precise agreement in the results from both counterbalanced studies of the WAIS-5 provides a degree of cross-validation of the present findings. However, the key analysis is the WAIS-IV vs. WAIS-5 study, which included a large heterogeneous sampling of adolescents and adults. Those findings generalize to older adolescents, young adults, middle-aged adults, and the elderly population. The results for 16-year-olds in the WISC-WAIS study generalize only to 16-year-olds. They do not generalize to adolescents in general, and they in no way suggest any reduction in the Flynn effect for children. The significant and striking interaction between age and the Flynn effect reported for 10,000+ adolescents ages 13–18 on a test of fluid reasoning ([31]) underscores the need to avoid generalizing the findings to any child or adolescent below age 16. [31] ([31]) observed the following Flynn effects for each separate age group in their large-scale investigation: age 13 (+2.33), age 14 (+0.88), age 15 (−0.89), age 16 (−0.19), age 17 (−0.76), and age 18 (−1.66).

## 4. Discussion

In scholarly works seeking to examine the “why” question of the Flynn effect, research has recognized the connection between environment and human growth intelligence (HGI), especially in the context of what environmental factors may have on both intellectual growth of people (HGI) and on technology (i.e., artificial general intelligence [AGI]; [24]). Various theories have tackled “why” from a theoretical perspective, such as through the co-occurrence model ([61]) and through the evaluation of several “Flynn paradoxes”. O’Keefe and colleagues (2023) argue two perspectives, speaking to the metaphor of “clouded spectacles” in understanding the reason for the effect, summarizing nicely the differing perspectives of (1) within-person and/or between-person changes or (2) implicit/explicit age/period/cohort effects ([29]). A robust list of potential reasons exist for the Flynn effect (see Table 1 in [29]), such as improvements across various arenas of health (e.g., nutrition, lead levels), education (e.g., better, longer, child/parental education), parenting styles (e.g., education, smaller number of children), social causes (e.g., niche selection, complex social settings), technology (e.g., computers, artificial light), genetic (e.g., heterosis), and others (e.g., slowed life, experience with tests, collective unconscious), with some of these concepts discussed at length in particular connection to children and adolescent cognitive development ([42]). Furthermore, scholars have suggested a “parental executive model”, which posits that adults involved in the lives of children consciously or unconsciously optimize outcomes through accessing and using the resources as mentioned above, which improves the child’s wellbeing and intellectual development ([35]). Given the fact that the pandemic intervened between the collection of “old test” data (WAIS-IV and WISC-V, data collected 2007–2008 and 2013, respectively) and “new” 2023 WAIS-5 test data, society’s broad response to the pandemic and its far-reaching impact into every aspect of American society introduces yet another intervening variable with an unknown (yet potentially powerful) interaction with the magnitude of the Flynn effect. Additionally, given the timely and recent discussion of the impact of social media on mental wellbeing and cognitive functioning, concepts related to modern day technological life may also become increasingly relevant ([20]).

One change from the WAIS-IV to WAIS-5 is noteworthy—the reduction in the number of subtests that contribute to the Full Scale, from 10 to 7. Compounding that change is the different contribution made by the primary indexes (and hence by the CHC broad abilities) to FSIQ. Most notably, the Processing Speed Index and Working Memory Index comprised 40% of the WAIS-IV Full Scale, a value that was reduced to 28.6% on the WAIS-5. Just as important, the construct measured by the WAIS-IV Working Memory Index differs from the comparable construct measured by the WAIS-5 because one subtest was eliminated (Arithmetic), one was added (Running Digits), and one was substantially modified (Digit Span, now called Digit Sequencing). There were also significant construct changes made to Digit Span, now called Digit Sequencing. Historically, each broad ability and each Wechsler subtest has a different history of Flynn effects ([18]). So, changing the mix of broad abilities and subtests that contribute to FSIQ could change the FE for FSIQ.

Regardless of “why”, the greatly reduced Flynn effect revealed in the analyses presented here, based on validity data reported in the WAIS-5 Technical and Interpretive Manual ([59]), has the potential to shake up the diagnostic process in clinics, schools, and courtrooms. As our study indicates, the Flynn effect for the American population of older adolescents and adults of all ages is no longer three points; rather, it is closer to 1 point. With evidence that the Flynn effect is substantially reduced, the AAIDD best practices for a three-point adjustment for outdated norms may need to be re-evaluated. Beyond providing an accurate representation of an individual’s abilities, this reduction in the Flynn effect may also impact who qualifies for state services that support those with an intellectual disability. Capital punishment cases may also be affected, strikingly so in many instances. With a reduced Flynn effect, the number of people who will no longer qualify for an intellectual disability will increase, ultimately meaning more deaths. We are in no way approving of the way IQ is being used to determine irrevocable life and death decisions, yet we understand clearly the potentially grave implications of such a finding. It is important to note that these findings are not generalizable to the entire population, as the sample was only for individuals aged 16 and older, which is the age range of people who potentially face the death penalty. As people first, none of the authors want more individuals to die; that is not the goal of this paper. Rather, our goal as scholars is to bring to light that the best science should be used to make decisions, especially high stakes decisions. With the recent publication of the WAIS-5, the gold standard and latest measure of intelligence, the present norming data suggest a different pattern of population IQ growth. With this, we ask, is it scientifically defensible to correct for IQ by three points given these most recent data with a strong sample, a counterbalanced design, and that were found post-pandemic, with arguably the best IQ measure on the market? Simply put, federal and state courts should review the present state of the literature to consider the reduced Flynn effect revealed in the analyses summarized here based on validity data reported in the WAIS-5 manual.

Table 2 illustrates how the Flynn effect can be a matter of life and death. This table illustrates test data for Jerome D. (a pseudonym). His sentencing hearing was held in 2024 in a state that does not permit the diagnosis of ID if a criminal’s IQ—after adjustment for the Flynn effect and practice effects—is 76 or greater. To meet the Prong 1 criterion for subaverage IQ in this state, the adjusted IQ must be 75 or below. Jerome D. was tested on the WISC-III in 1991 as a 16-year-old while serving in a juvenile detention center. He was subsequently administered the WAIS-III at age 25 and again at age 30 within the prison system. After his release, he was convicted of murder in 2009, administered the WAIS-IV, and sentenced to be executed. He was on death row for years and was ultimately granted a new sentencing hearing in 2024 as an Atkins case. He was once again administered the WAIS-IV, this time when the norms were extremely out of date. When his adjusted IQs are examined over time, he never scored below 76 on any assessment, although he came close every time—until he was administered the very out-of-date WAIS-IV. The Flynn effect mandated subtracting 5 points from his obtained IQ of 80, and the 75 qualified him for an ID diagnosis when paired with Prong 2 (subaverage adaptive behavior) and Prong 3 (early onset) criteria, both of which were met. But how different the outcome would have been with a Flynn effect of 1.2 points per decade! The adjustment would have been 2 points, and his adjusted IQ would have been 78—too high for an ID diagnosis. These issues must be weighed carefully by AAIDD, DSM-5-TR, and the court system to decide the most appropriate guidelines for ID diagnosis as we move toward the future.

And the Flynn effect reduction is not the only provocative finding presented by Wechsler et al. 2024. Additionally, the test–retest reliability study (*N* = 201, ages 16–90, *M* = 53, mean interval = 29 days) reveals that the practice effect (see Table 3)—like the Flynn effect—has gotten smaller over time. Table 3 shows mean gain scores from test to retest on three versions of WAIS for different age groups ([54], [57]; [59]). For the WAIS-III, mean gains on FSIQ were about 4–6 points for young and middle-aged adults; for the WAIS-IV, those gains were 4.5–5 points; but on the WAIS-5, they dropped to 3.7 IQ points for ages 16–34 and 35–69. The fact that the practice effect has historically been smaller for elderly adults than for young and middle-aged adults was maintained for the WAIS-5, but again the differences across generations are notable. FSIQ gain scores for the elderly were 3–4 points for both the WAIS-III and WAIS-IV. That number dipped to 2.3 points for the WAIS-5 at ages 70–90. 

Just as the Flynn effect requires careful consideration when diagnosing ID, so too does the practice effect. Whereas the specific numerical adjustment is quantified for the Flynn effect (3 points per decade), it is not for the practice effect. Nonetheless, both the AAIDD diagnostic manual ([40]) and DSM-5-TR ([1]) make it imperative that examiners take practice effects fully into account when diagnosing ID.

It is common for incarcerated people on death row to have multiple Wechsler’s scores in their records, often four or more. There are childhood measures (e.g., Wechsler Intelligence Scale for Children [WISC]) from special education and juvenile court and WAIS measures from earlier imprisonments prior to the crime, which resulted in a death sentence. Sometimes it is the practice effect that holds the clue to a person’s “true IQ”.

Table 4 shows the repeated Wechsler tests administered to a death row incarcerated individual, Mr. Justin C. (a pseudonym). He was tested seven times between ages 10.5 and 32. He was initially given the WISC-R (Wechsler Intelligence Scale for Children, Revised) when referred for special education in fourth grade, and was given the same test at ages 11 and 12.5 by juvenile court psychologists. This pattern continued such that when Mr. C. was arrested in 1997 for murder (for which he received the death penalty), he had been tested on WISCs five times before achieving his first WAIS. He was on death row for 12 years when his lawyers appealed his sentencing claiming the “Atkins defense” of ID. He was administered the WAIS-IV, which had only been out for a year. So, the Flynn effect was not an issue, and neither was the practice effect because of the 12-year interval since the previous Wechsler administration.

Table 3 shows exactly how the practice effect (known as progressive error when multiple Wechsler scales are administered to the same person) can impinge on accurate diagnosis. Nonverbal and speeded tasks have a substantially larger practice effect than verbal tasks. Traditional performance tests are intended to measure fluid reasoning and visual-spatial problem solving, the types of problems not taught in education. When these tasks are administered a second time, the novelty is gone, and nonverbal standard scores are spuriously inflated by 7 to 10 points (verbal tasks go up by 2–3 points); Full Scale IQs rise by 5–8 points. When a person is tested a third time, a fourth time, and so forth, the progressive error continues to inflate the person’s nonverbal and global scores.

Mr. C.’s scores over time illustrate these phenomena. His initial Performance IQ of 74 was untainted. But the 86 he earned nine months later was exactly what you would predict from the practice effect, and the values in the 80s and 90s are vintage progressive errors. Switching from the WISC to the WAIS at age 20 makes no difference. The practice effect holds regardless of which Wechsler scale is administered.

Therefore, all Full Scale IQs obtained at ages 11 through 20 (mostly in the 80s) are spuriously high due to the practice effect and progressive error and are not interpretable. The FSIQ of 72 at age 10.5 would be adjusted downward into the 60s because of the Flynn effect (the WISC-R norms were 16 years old when Mr. C. was tested). What is Mr. C.’s true IQ? This is clearly not a simple question. Can one have confidence that his WAIS-IV IQ of 73 is valid? No, because faking a bad score is always a possibility for a person on death row. But clearly, practice effects can play a key role in capital punishment cases, no less than the Flynn effect. In sum, given the recent findings in the WAIS-5 (a strong sample with counterbalanced design on the gold standard most up to date measure) should be considered in light of other findings highlighting the suspected diminishing Flynn effect by decision-makers involved in the AAIDD and DSM-5-TR for ID diagnosis, and by the courts for capital punishment cases.

## 5. Limitations and Future Directions

### 5.1. Limitations

As noted, the results of the WISC-V vs. WAIS-5 do not generalize to any other age group except 16-year-olds. We are left with no knowledge of the contemporary Flynn effect for anyone ages 15 and younger. This limitation is frustrating given the precedent of prior research on the impact of the Flynn effect in younger populations of adolescents and children ([31]) and the documented decline of reading and mathematics abilities across the United States ([49]). Findings from the National Assessment of Educational Progress (NAEP) indicate a stark decline in reading (5-point drop) and mathematics (7-point drop) scores for 9-year-old children during the COVID-19 pandemic, the largest drop in performance since 1990 ([49]). This decline is not new. Decreases in testing scores have been documented for students before the pandemic, too ([50]). Prior to the pandemic, reading scores for 4th grade, 8th grade, and 12th grade students had decreased by 1 point, 3 points, and 2 points from 2015 to 2019, respectively ([50]). It is clear that these decreases in performance lead to a future call for research in this area. Undoubtedly, that future call will be answered by Pearson when the WISC-6 is published in a few years.

Another limitation is that counterbalanced studies necessarily use that experimental design to control for the practice effect. That means whichever test is administered second (whether the older test or the newer test) will have an added bump from the practice effect. The ultimate mean difference between the old test and the new test is the average of the mean IQs obtained from each test sequence—e.g., WAIS-IV first—WAIS-5 second and WAIS-5 first—WAIS-IV second [59] ([59]) explain: “For the counterbalanced studies, means, *SD*s, and correlation coefficients corrected for range restriction were calculated separately for the portion of the sample taking each administration sequence and then averaged. This method prevents repeated administration effects from artificially lowering the correlation coefficients” (p. 83). The statistical procedure is the correct one for dealing with the impact of practice effects. But counterbalanced designs, by definition, introduce unwanted errors into the analyses. A second important limitation of our study is that data are based on relatively small sample sizes—<200 for one study and <100 for the second study. These sample sizes are ample for producing results and corresponding implications with evidence of reliability and validity. But, nonetheless, the Flynn effect is intended to provide a broad-based comparison of large normative samples (each >2000); and it does this by analyzing data from much smaller subsamples.

### 5.2. Future Directions

A natural curiosity given the limitations of the present data set for age includes seeing if this pattern is observed to stand with younger adolescents and children. With the sixth edition of the slotted Wechsler Intelligence Scale for Children (WISC-6) for release within the coming years (it is currently in the pilot testing phase), the examination of these findings in a more robust sample will be helpful to more fruitfully understand the connection between adult findings and youth implications. Additionally, as scholars, we are left wondering about the following: how do these effects hold up on a subtest level? Several researchers have expressed interest in understanding the shift in IQ to a domain level interpretation as opposed to a Full Scale measure ([5]).

Additionally, future research on the question of how construct and subtest changes from the WAIS-IV to the WAIS-5 may have affected the Flynn effect for adults is needed. These modifications to the Full Scale are an alternative hypothesis that competes with the hypothesis that the Flynn effect is in decline. At the very least, these changes may explain a portion of the apparent decline.

Finally, the authors are curious to understand the “why” of these findings in a modern lens. Given the resounding research and clinical interest in social media on development ([20]), coupled with the impact of a pandemic on a developing brain, the “why” within a more modern approach to consider the cognitive impact of social media, technology, and a worldwide pandemic is of interest ([20]). Finally, this subject will be of great interest to the AGI development world to understand how changes in human general intelligence across generations can, and might, impact artificial general intelligence for bots.

## Figures and Tables

**Table 1 jintelligence-12-00118-t001:** Flynn effects on Wechsler scales for ages 16–90 years based on counterbalanced studies of WAIS-III, WAIS-IV, WAIS-5, and WISC-V.

WAIS Versions	*N*	Age Range (Mean)	Mean FSIQ Older Wechsler Scale	Mean FSIQ Newer Wechsler Scale	Mean IQ Difference	Points/Decade (Flynn Effect)
WAIS-III vs. WAIS-IV						
Non-Clinical	240	16–88 (52.7)	102.9	100.0	2.9	2.6
Low-IQ	49	16–65 (30.6)	68.6	65.5	3.1	2.8
WAIS-IV vs. WAIS-5						
Non-Clinical	186	16–90 (47.8)	101.6	99.7	1.9	1.2
WAIS-5 vs. WISC-V						
	98	16-0–16-11 (16.4)	100.4	99.2	1.2	1.2

**Table 2 jintelligence-12-00118-t002:** Wechsler Full Scale IQs earned by Jerome D. between 1991 and 2024, starting in juvenile detention and continuing on death row.

Test	WISC-III	WAIS-III	WAIS-III	WAIS-IV	WAIS-IV
Year	1991	2000	2005	2009	2024
Age	16	25	30	34	49
IQ	77	78	81	78	80
How Outdated Are Norms?	2 years	5 years	10 years	1.5 years	16.5 years
Adjusted IQ	(0.6 points) 76.4	(1.5 points) 76.5	(5 points) 78	(0.5 points) 77.5	(5 points) 75

**Table 3 jintelligence-12-00118-t003:** Practice effects on Full Scale IQ, by age (WAIS-III to WAIS-IV to WAIS-5).

WAIS Version Year	Mean Gain (Younger Age)	Mean Gain (Middle Age)	Mean Gain (Middle Age)	Mean Gain (Older Age)
WAIS-III	16–29 (+5.7)	30–54 (+5.1)	55–74 (+3.9)	75–89 (+3.2)
WAIS-IV	16–29 (+4.6)	30–54 (+4.4)	55–69 (+5.0)	70–90 (+3.9)
WAIS-5	16–34 (+3.7)	35–69 (+3.7)	–	70–90 (+2.3)

Note. Data are from *WAIS-IV Technical & Interpretive Manual* ([57]) and from *WAIS-5 Technical & Interpretive Manual* ([59]).

**Table 4 jintelligence-12-00118-t004:** Justin C. sample Flynn effect case.

Test Date	Test	Justin’s Age	Verbal IQ	Performance IQ	Full Scale IQ
February 1988	WISC-R	10 ½	73	74	72
November 1988	WISC-R	11	70	86	76
April 1990	WISC-R	12 ½	81	96	87
March 1992	WISC-III	14 ½	77	90	81
December 1993	WAIS-IV	16	70	92	80
November 1997	WAIS-IV	20	75	98	85
December 2009	WAIS-IV	32	72	77	73

Note. Data are from WAIS-IV Technical & Interpretive Manual ([57]) and from WAIS-5 Technical & Interpretive Manual ([59]). Low IQ comprise 25 adults diagnosed with mild intellectual disabilities + 24 with borderline intellectual functioning. Test intervals averaged about 1 month.

## Data Availability

Data are available in the WAIS-5 Technical and Interpretive Manual ([59]).

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
