# Peer review of "Wait, Where’s the Flynn Effect on the WAIS-5?"

_jintelligence, 2024, doi:10.3390/jintelligence12110118_

Round 1

Reviewer 1 Report

Comments and Suggestions for Authors

Thank you for the opportunity to review this manuscript.

The authors have anlayzed the decline in Flynn effect based on the comparison of mean IQ scores on the WAIS-IV and WAIS-5 tests. The tests were administered to relatively small samples (<200 for one study and <100 for the second study). In the results section on page 5 and in the limitations section on page 9 the authors mention that the results only generalize to 16-year-olds. The authors also note that the tests exhibit a decline in practice effect when multiple Weschler scales are administered to the same person. The tests are unable to inform us how the Flynn effect has changed for the younger adolescents and children.

Despite these limitations, the authors argue that these findings should draw the immediate attention of the court, with the potential that the court awarded deaths may increase if the Flynn effect adjustments are made on the reduced per decade adjustment of about 1 point per decade rather than 3 points per decade. If the findings of this study could potentially lead to an increase in death sentences, and if the reduced Flynn effect is not yet generalizable to the full population, why should the court consider the findings of this study with urgency? Wouldn’t the court want to wait for more generalizable and convincing evidence than the weakness of the study regarding small sample sizes and lack of generalizability across all ages? Aren’t the authors overselling their findings?

The authors may consider toning down their emphasis about the urgency with courts and lay out the limitations of the study in the abstract and introduction rather than later.

There is not much data in the paper on how clinics and schools would be affected by the findings of this study. Can the authors cite data on how many cases in clinics, schools, and courtrooms, have been based on the old Flynn effect since the pandemic? Empirical data is needed to support the author's concerns. The paper can benefit from a review of how clinics and schools base policies on the use of the Flynn effect. Appropriate literature should be cited.

The authors could add some discussion on the decline in test scores on the two tests related to the National Assessment of Educational Progress (main NAEP and (Long Term Trend) LTT NAEP). The post pandemic decline in test scores has been observed in both NAEP and LTT NAEP for students without and with disabilities. The test score decline post-pandemic has been observed internationally. The authors can cite several papers here (look at the works by Eric Hanushek, Ludger Woessmann, Harry Patrions, the education next magazine, NBER website, and so forth).

Some data on pre-and post-pandemic identification on Intellectual Disability cases in schools and among adults could be added.

The authors mention Hadit on the negative effects of the role of social media since 2012 on the population. This discussion may be presented upfront rather than at the end of the paper.

As the authors note on page 2, Flynn himself hypothesized that the effect would decline over time.

The authors may cite this paper in their review of the literature https://link.springer.com/article/10.1007/s10648-021-09657-y

Author Response

Comments 1: Despite these limitations, the authors argue that these findings should draw the immediate attention of the court, with the potential that the court awarded deaths may increase if the Flynn effect adjustments are made on the reduced per decade adjustment of about 1 point per decade rather than 3 points per decade. If the findings of this study could potentially lead to an increase in death sentences, and if the reduced Flynn effect is not yet generalizable to the full population, why should the court consider the findings of this study with urgency? Wouldn’t the court want to wait for more generalizable and convincing evidence than the weakness of the study regarding small sample sizes and lack of generalizability across all ages? Aren’t the authors overselling their findings?

Response 1: More clarification was added to help describe the function of the paper and not overstating the findings, while also paying homage to what this means for scientifically informed practice while also considering humanity. See the discussion section: “It is important to note that these findings are not generalizable to the entire population, as the sample was only for individuals aged 16 and older, which is the age range of people who face potential death penalty. As people first, none of the authors want more indi-viduals to die, that is not the goal of this paper. Rather, our goal as scholars is to bring to light that the best science should be used to make decisions, especially high stakes de-cisions. With the recent publication of the WAIS-5, the gold standard and latest measure of intelligence, the present norming data suggest a different pattern of population IQ growth. With this, we ask, is it scientifically defensible to correct for IQ by three points given these most recent data with a strong sample, counter balanced design, post-pandemic, with arguably the best IQ measure on the market? Simply put, federal and state courts should review the present state of the literature to consider the reduced Flynn effect revealed in the analyses summarized here based on validity data reported in the WAIS-5 manual.”

Comments 2: The authors may consider toning down their emphasis about the urgency with courts and lay out the limitations of the study in the abstract and introduction rather than later. 

Response 2: Added to the abstract: Limitations include lack of generalization to children, impact of the practice effects, and a small sample size. Added to the Introduction (page 2): Even within counterbalanced research (such as within the present paper), concerns with sample size and practice effects as well as generalizability (such as with differing age groups), may come into effect. Language softened throughout (see pages 6-8).

Comments 3: There is not much data in the paper on how clinics and schools would be affected by the findings of this study. Can the authors cite data on how many cases in clinics, schools, and courtrooms, have been based on the old Flynn effect since the pandemic? Empirical data is needed to support the author's concerns. The paper can benefit from a review of how clinics and schools base policies on the use of the Flynn effect. Appropriate literature should be cited.

Response 3: Whereas these results may impact diagnosis in high school there are no generalization or implications that findings are relevant to children or adolescents up to age 16. With an average age of 47, this is a study about adults, about the adult Flynn effect, not addressed to the school systems.

However, given these comments, we also have added additional information regarding clinical and school practice. This includes implications of using adult cognitive assessments with outdated norms for students who have qualified for special education under the intellectual disability category. This can be found in the Why does the Flynn Effect Matter? Secion: “In the school and clinical settings, the Flynn effect adjustment is less relevant. The McGrew (2015) states, “The use of the Flynn effect correction in clinical settings is less of an issue given that psychologists in such settings typically have more leeway to interpret scores as ranges, invoke clinical judgment, and incorporate information regarding measurement error in interpretation of scores when making a diagnosis” (p. 160). School psychologists follow similar guidelines within the school setting. Additionally, in schools, the Flynn effect is less relevant due to the evaluation mandates put forth by the Individuals with Disabilities Education Act (IDEA 2004). Students who qualify for special education are required to be re-evaluated at least once every three years (U.S. Department of education 2004). However, for students with an intellectual disability who are transitioning out of high school (i.e., over the age of 16), obsolete tests and outdated norms for adult cognitive assessment tools can lead to inflated IQ scores which may change eligibility classification from ID to learning disability (Trahan et al. 2014). For students who are leaving high school and entering the independence required of adulthood, this change in classification may impact government supported social security and disability supports that require evidence of impaired intellectual functioning (Trahan et al. 2014).”

Comment 4: The authors could add some discussion on the decline in test scores on the two tests related to the National Assessment of Educational Progress (main NAEP and (Long Term Trend) LTT NAEP). The post pandemic decline in test scores has been observed in both NAEP and LTT NAEP for students without and with disabilities. The test score decline post-pandemic has been observed internationally. The authors can cite several papers here (look at the works by Eric Hanushek, Ludger Woessmann, Harry Patrions, the education next magazine, NBER website, and so forth).

Response 4: We thank you for this suggestion. We have added information on NAEP to our future directions section of our paper. We believe these scores in addition to our note of the importance of the WISC-6 publication in a few years emphasizes the importance and need to research in this area further.

Comments 5: Some data on pre-and post-pandemic identification on Intellectual Disability cases in schools and among adults could be added.

Response 5: Thank you for this suggestion. We spent some time looking for data on pre and post pandemic and were unable to find any additional studies to address this point at this time.

Comments 6: The authors mention Haidt on the negative effects of the role of social media since 2012 on the population. This discussion may be presented upfront rather than at the end of the paper.

Response 6: A mention to Haidt’s work was added on page 6: “Additionally, given the timely and recent discussion of the impact of social media on mental wellbeing and cognitive functioning, concepts related to modern day techno-logical life may also become increasingly relevant (Haidt, 2024).”

Comments 7: As the authors note on page 2, Flynn himself hypothesized that the effect would decline over time.

Response 7: We agree, we added in a sentence to highlight the findings of this paper to align with his hypothesis: “Recent research, along with findings from this paper, highlight that his original hypothesis may indeed be correct.”

Comments 8: The authors may cite this paper in their review of the literature https://link.springer.com/article/10.1007/s10648-021-09657-y

Response 8: This article was added in two reference points, one addressing the limitations of the prior research with proxy for IQ (see page 2) and the other speaking to the connection of the concept to children (see page 6). Thank you for sharing this resource with us.

Reviewer 2 Report

Comments and Suggestions for Authors

This article raises several important questions the field must grapple with.  However, it provides no new data as all these data have already been reported in the test manual, and no new anlayses of the previously reported data.

Table 1 appears to show a reverse FE for 16 year olds, which is not discussed in the text.  Is the table incorrect?

It is odd to discuss Table 4 before Table 3 in the text.

It is known that some broad factors / subtests have more or less Flynn and practice effects.  The paper should mention changes in the composition of FSIQ (reduced from 10 to 7 subtests) among the possible reasons for the reduced Flynn and practice effects.  Perhaps the finding is not so surprising when the new composition of FSIQ is considered (reduced contribution of working memory and processing speed subtests)? 

Author Response

Comments 1: This article raises several important questions the field must grapple with.  However, it provides no new data as all these data have already been reported in the test manual, and no new analyses of the previously reported data.

Response 1: We hope to make clearer that the information we are presenting is not reported in the manual, it provides additional calculations based on the manual. We added in the following to the Procedure section: “This particular paper reports additional Mean IQ Difference, not reported in the manual, which has implications for understanding the Flynn effect between the WAIS-IV, WISC-V, and WAIS-5.”

Comments 2: Table 1 appears to show a reverse FE for 16 year olds, which is not discussed in the text.  Is the table incorrect?

Response 2: Thank you for this feedback. This error was corrected, thank you for catching this error.

Comments 3: It is odd to discuss Table 4 before Table 3 in the text.

Response 3: We were unsure if this comment referred to the mention of the tables or the placement of the tables, but to be sure, we will address both! Table 3 is discussed on page 7, Table 4 is discussed on page 8, so that should be clear. We moved formatting around to have Table 3 appear higher up in the manuscript if that eases the reading

Comments 4: It is known that some broad factors / subtests have more or less Flynn and practice effects.  The paper should mention changes in the composition of FSIQ (reduced from 10 to 7 subtests) among the possible reasons for the reduced Flynn and practice effects.  Perhaps the finding is not so surprising when the new composition of FSIQ is considered (reduced contribution of working memory and processing speed subtests)? 

Response 4: In order address this comment, a paragraph was added to the discussion: “Broad factors and subtests can also be subject to the Flynn effect and practice effects. With the structural changes from the WAIS-IV to the WAIS-5, it is wise to wonder: are the changes in composition of Full Scale IQ (reducing from ten to seven subtests) among potential rationale for the observed changed in the Flynn effect and practice effects? In general, the average effect gain across the subtests was +0.43 on the WAIS-5 with the seven-subtest model. The three remaining subtests on the WAIS-IV demonstrate an increase of +0.40. Thus, when looking at the seven retained subtests and the three not included in the Full Scale score, the average Flynn effect is similar, thus likely not a factor in the comparison from the WISC-V to WAIS-V both use seven subtests, so this is less likely an issue.”

Round 2

Reviewer 1 Report

Comments and Suggestions for Authors

The article can be published.

Author Response

Thank you so much for the support of the publication of this piece. There were no comments made by the reviewer, excepting "The article can be published." Thank you. 

Reviewer 2 Report

Comments and Suggestions for Authors

Lines 311 - 316 attempt to address comment 4 from the first round.  However, these lines are difficult to follow and unconvincing.  First, I don't understand how the average subtest effect in the 7 subtest model can equal 0.43 when the FSIQ effect is 1.9 points (7 x .43 does not equal 1.9).  Second, it's unclear which other three WAIS-4 subtests average .40 points given that there are new subtests in the 10 subtest model, such as Running Digits and Figure Weights.  Also, there is no mention of the significant construct changes made to Digit Span, now called Digit Sequencing.  Moreover, the issue is not the number of subtests but rather the changing percentages of broad abilities contributing to FSIQ.  Historically, each broad ability has a different history of Flynn effects.  So, changing the mix of broad abilities that contribute to FSIQ could change the FE for FSIQ.  For example, the contribution of processing speed and working memory subtests to FSIQ was reduced from 40% in WAIS4 (4 of 10 subtests) to 28% in WAIS5 (2 of 7 subtests).  

I realize that this paper is focused on the need to adjust FSIQ in high stakes decisions, and not on understanding the reasons for the reduced effect.  So, perhaps just call for future research on the question of how construct changes may have effected the FE.  After all, this is an alternative hypothesis that competes with the hypothesis that FE is in decline.  

Overall, the orignal contribution of this research is limited to subtracting the old and new FSIQ means already reported in the manual.  It is more of an opinion paper than a research study.

Line 75; It is unclear if "his original hypothesis" refers to Flynn's prediction of 3 points per decade or his 2007 warning that the effect may not last forever.

Author Response

Comment 1: Lines 311 - 316 attempt to address comment 4 from the first round.  However, these lines are difficult to follow and unconvincing.  First, I don't understand how the average subtest effect in the 7 subtest model can equal 0.43 when the FSIQ effect is 1.9 points (7 x .43 does not equal 1.9).  Second, it's unclear which other three WAIS-4 subtests average .40 points given that there are new subtests in the 10 subtest model, such as Running Digits and Figure Weights.  Also, there is no mention of the significant construct changes made to Digit Span, now called Digit Sequencing.  Moreover, the issue is not the number of subtests but rather the changing percentages of broad abilities contributing to FSIQ.  Historically, each broad ability has a different history of Flynn effects.  So, changing the mix of broad abilities that contribute to FSIQ could change the FE for FSIQ.  For example, the contribution of processing speed and working memory subtests to FSIQ was reduced from 40% in WAIS4 (4 of 10 subtests) to 28% in WAIS5 (2 of 7 subtests).   

Response 1: Thank you so much for your insight. We completely agree, we did not adequately address Comment 4 from the first round. We have eliminated the confusing statistical analysis and have replaced that entire section with the following: “One change from the WAIS-IV to WAIS-5 is noteworthy—the reduction in the number of subtests that contribute to the Full Scale, from 10 to 7. Compounding that change is the different contribution made by the primary indexes (and hence by the CHC broad abilities) to FSIQ. Most notably, the Processing Speed Index and Working Memory Index comprised 40% of the WAIS-IV Full Scale, a value that was reduced to 28.6% on the WAIS-5. Just as important, the construct measured by WAIS-IV Working Memory Index differs from the comparable construct measured by the WAIS-5 because one subtest was eliminated (Arithmetic), one was added (Running Digits), and one was substantially modified (Digit Span, now called Digit Sequencing). There also were significant construct changes made to Digit Span, now called Digit Sequencing.  Historically, each broad ability and each Wechsler subtest has a different history of Flynn effects (Flynn & Weiss, 2007). So, changing the mix of broad abilities and subtests that contribute to FSIQ could change the FE for FSIQ.” 

Comment 2: I realize that this paper is focused on the need to adjust FSIQ in high stakes decisions, and not on understanding the reasons for the reduced effect.  So, perhaps just call for future research on the question of how construct changes may have affected the FE.  After all, this is an alternative hypothesis that competes with the hypothesis that FE is in decline.   

Response 2: We added a future direction given this information: “Additionally, future research on the question of how construct and subtest changes from the WAIS-IV to the WAIS-5 may have affected the Flynn effect for adults is needed. These modifications to the Full Scale are an alternative hypothesis that competes with the hypothesis that Flynn effect is in decline. At the least, these changes may explain a portion of the apparent decline.” We did keep in reference to the “why” because although we did not address it in this paper, we do feel this is important for readers to be aware of and engage in their own respective inquiry or follow up on those items. 

Comment 3: Overall, the original contribution of this research is limited to subtracting the old and new FSIQ means already reported in the manual.  It is more of an opinion paper than a research study. 

Response 3: The paper has been revised in the last round to be an opinion paper.  

Comment 4: Line 75; It is unclear if "his original hypothesis" refers to Flynn's prediction of 3 points per decade or his 2007 warning that the effect may not last forever. 

Response 4: Thank you for the feedback. We changed the section to read “highlight that Flynn’s original hypothesis may indeed be correct that the effects may not last forever.” 

Round 3

Reviewer 2 Report

Comments and Suggestions for Authors

Thank you for addressing my concerns.  As an opinion piece, I withdraw my objection that there is no new data presented.  It is likely that this paper will be introduced as evidence into many capital court cases.